# Live Viral Vaccine Neurovirulence Screening: Current and Future Models

**DOI:** 10.3390/vaccines9070710

**Published:** 2021-06-30

**Authors:** Corey May Fulton, Wendy J. Bailey

**Affiliations:** Safety Assessment and Laboratory Animal Resources, Merck & Co., Inc., West Point, PA 19486, USA; wendy_bailey@merck.com

**Keywords:** live viral vaccines, vaccine safety, neurovirulence, monkey neurovirulence test, live attenuated vaccines

## Abstract

Live viral vaccines are one of the most successful methods for controlling viral infections but require strong evidence to indicate that they are properly attenuated. Screening for residual neurovirulence is an important aspect for live viral vaccines against potentially neurovirulent diseases. Approximately half of all emerging viral diseases have neurological effects, so testing of future vaccines will need to be rapid and accurate. The current method, the monkey neurovirulence test (MNVT), shows limited translatability for human diseases and does not account for different viral pathogenic mechanisms. This review discusses the MNVT and potential alternative models, including in vivo and in vitro methods. The advantages and disadvantages of these methods are discussed, and there are promising data indicating high levels of translatability. There is a need to investigate these models more thoroughly and to devise more accurate and rapid alternatives to the MNVT.

## 1. Introduction

Live viral vaccines are among the most successful preventive measures used to control viral spread and disease in the world. They were used to eradicate smallpox in 1980, and they have been critical to efforts to control or eliminate widely spread, debilitating viral diseases. Many of these viruses, such as measles, mumps, and polio, have been associated with severe neurological disease. Approximately half of all emerging viruses cause severe encephalitis or serious neurological complications [1], including SARS-CoV-2 [2,3,4,5]. Unlike most other acute health effects from viral infection, neurological damage can be long-term and life-altering for patients. Because of this, there is a critical need for vaccine development to be rapid without compromising safety. It is essential to ensure a live viral vaccine is free of detrimental neurological effects early in development. We must explore novel approaches to meet these conflicting requirements: producing safe, effective vaccines in record time.

There are two major factors that contribute to viral damage to the central nervous system (CNS): neuroinvasion and neurovirulence. Neuroinvasion is the ability of a virus to enter the CNS through the intrinsic barriers that separate the spinal cord and brain from the peripheral regions and bloodstream of an organism. Neurovirulence is the ability of a virus to cause damage to the cells of the CNS after the virus has invaded, which ultimately leads to clinical outcomes. This review will focus on testing for neurovirulence in live viral vaccines derived from neurovirulent wild-type strains, the testing that is required to determine if neurovirulence remains or recurs in live viral vaccines, and potential safety models for testing vaccines in the future.

## 2. Live Viral Vaccines and Neurovirulence

### 2.1. Neurovirulence

It is important to define which viruses are and are not neurovirulent for vaccine development purposes. Many viruses can cause damage to the CNS through direct and/or indirect mechanisms (e.g., direct infection of neurons or infiltration of peripheral inflammatory cells, respectively), so definitions can be subjective. Some viruses are clearly neurovirulent. Poliovirus infection causes long-lasting paralysis through damage to the spinal cord, brainstem, and cortex in approximately 1–2% of patients [6]. West Nile virus and Japanese encephalitis virus show similar incidence of CNS involvement in patients as poliovirus [7,8]. The neurotropic alphaviruses Western equine encephalitis, Eastern equine encephalitis, and Venezuelan equine encephalitis cause neurological involvement in at least 5% of infections [9,10,11]. For other viruses, the incidence of neurological disease is lower. Measles is considered a neurovirulent virus, but its primary pathogenesis is in the respiratory tract. Measles virus infection of the CNS is rare but severe and appears to require significant mutations in the viral genome. The neurological effects of measles virus infection occur either as subacute sclerosing panencephalitis (SSPE) or measles inclusion body encephalitis (MIBE) [12]. MIBE occurs in immunocompromised patients, while SSPE can occur in any patient. Viruses isolated from patients with SSPE and MIBE have significant mutations to the genome that make them phenotypically distinct from wild-type measles. SSPE occurs in approximately 4–11 of every 100,000 measles infections [13]. Both MIBE and SSPE are associated with high levels of mortality, though this is months to years after initial infection [14]. A major factor that limits the ability to study the neurovirulence of measles virus is that these variants do not form cell-free virions [15,16,17]. The inability to isolate cell-free virus makes it difficult to study these specific, rare, and neurovirulent variants in any system. Given the variability between viruses and clinical outcomes, it is difficult to develop firm guidance as to which viruses should be considered neurovirulent for regulatory purposes. There is a wide range in clinical signs, time to showing these signs, and incidence of neurological involvement between viruses considered neurovirulent, which leads to inconsistency in regulatory controls for vaccines.

### 2.2. Live Viral Vaccines

Live viral vaccines (including chimeric, viral-vectored, and attenuated vaccines) offer many advantages over other types of vaccines. Live viral vaccines replicate within the host, increasing the amount of antigen produced with less inoculum necessary. In this way, they closely mimic natural infection, stimulating multiple arms of the immune system by directing the appropriate immune response to a specific virus. Live viral vaccines tend to elicit longer lasting immunity than subunit or killed vaccines, and usually only require a single administration. Killed and subunit vaccines often require the addition of substances to stimulate the immune response, known as adjuvants. Because live viral vaccines actively replicate, they generally do not require adjuvants. A review covering the history of live viral vaccines, as well as the strengths and hurdles of their development, was written by Phillip Minor [18]. Minor’s review highlights that the most commonly used vaccines were developed through random mutation rather than rational design, and how this approach does not work for many of the most prominent viral agents today.

The most widely used vaccines are primarily live viral vaccines. All of the components of the measles, mumps, rubella, and varicella vaccine are live viral vaccines. The yellow fever vaccine 17D (YFV 17D) and Sabin strain of poliovirus vaccine are also live viral vaccines that have been used to curb the spread and effects of both of those viruses. Live viral vaccines continue to be developed. The vaccine against Ebola virus credited with stopping the 2014–2015 outbreak is a chimeric live viral vaccine [19], as are the Johnson and Johnson and AstraZeneca vaccines against SARS-CoV-2 [20].

### 2.3. Attenuation of Neurovirulence in Live Viral Vaccines

Live viral vaccines need to be confirmed to be properly attenuated, as underattenuated vaccines may pose a risk of severe adverse events in patients that could reduce public trust in the safety of vaccines. Global health authorities such as the World Health Organization (WHO) provide guidance on neurovirulence testing [21]. For viruses that can potentially infect the CNS, this includes confirmation of absence of neurovirulence. The most commonly used live viral vaccines, including YFV 17D; the measles, mumps, rubella, and varicella vaccines; and the Sabin strain of poliovirus vaccine were developed through serial passage of wild-type viruses through cell culture, eggs, or animal models, accumulating mutations until they became attenuated. Notably, this method means that the mechanisms of attenuation for these live viral vaccines are often not well characterized. The lack of knowledge about specific mechanisms of attenuation makes development of future live attenuated vaccines more difficult, as the same attenuating mutations cannot be reproduced as quickly in new vaccines.

More recent live viral vaccines have been developed either through planned mutations in wild-type viruses using reverse genetics or through development of chimeric viruses. Targeted mutations were used in a Venezuelan equine encephalitis virus vaccine to develop a live attenuated vaccine to replace existing TC83 that shows promise as being protective and non-neurovirulent [22,23]. Chimeric vaccines have also shown promise for controlling dangerous viruses. For example, the vaccine against Ebola virus is a chimera using the Ebola glycoprotein to replace the glycoprotein in the vesicular stomatitis virus (VSV) genome. This vaccine was shown to lack neurovirulence compared to infections with VSV in a monkey model [24]. Chimeric vaccines using the YFV 17D genome as a backbone have been developed and tested for efficacy and lack of neurovirulence as well, replacing the envelope and pre-membrane proteins with target viruses. Some examples include vaccines against West Nile virus, Japanese encephalitis virus, Zika virus, and dengue viruses have been tested for neurovirulence and shown significant attenuation [25,26,27,28,29,30].

For any of the methods used to attenuate live viruses for use as vaccines, it is important to confirm the lack of neurovirulence during preclinical testing. As of now, testing is widely performed to confirm lack of neurovirulence, but the only test consistently used is the monkey neurovirulence test.

### 2.4. The Existing Test for Neurovirulence: The Monkey Neurovirulence Test

Different regulatory agencies have different requirements for proving the lack of neurovirulence of live viral vaccines, but all require the use of the monkey neurovirulence test (MNVT) for screening. The MNVT is considered the gold standard for neurovirulence screening and is used for both preclinical assessment of novel viral vaccines and for lot release testing of select existing viral vaccines. The procedure for the MNVT was first described in 1943 by Fox and Penna [31] in response to lot-to-lot variability in the YFV 17D that resulted in vaccine-associated neurological adverse events in patients [32]. Different strains of YFV 17D vaccine were inoculated intracranially into rhesus macaques, which were then assessed for clinical signs of encephalitis. Lots associated with encephalitis in patients showed clinical signs of encephalitis in the monkey model. In 1987, The MNVT was revised to include histopathological evaluations of specific brain regions rather than observed clinical, neurological changes. This focused on different regions of the brain as target regions, discriminator regions, and unaffected regions [33]. Target regions showed inflammation when infected with neurovirulent and non-neurovirulent strains, while discriminator regions showed significantly more inflammation when infected with neurovirulent strains. Current interpretation uses histopathology and measures glial activation and infiltration of peripheral immune cells into the CNS, which is semi-quantitatively scored. This scoring is based on histopathological changes but does not use specific cell markers.

The MNVT demonstrated clinical translational capabilities for live poliovirus vaccine screening. The MNVT was used for lot release testing of Sabin live attenuated poliovirus vaccine [34]. The method was changed to intraspinal inoculation, as this appears to improve sensitivity for the poliovirus vaccine [35]. The MNVT is required for lot release testing of each strain of the poliovirus vaccine by the WHO, Japanese Pharmaceuticals and Medical Device Agency (PMDA), US Food and Drug Administration (FDA), and European Medicines Agency (EMA). With the exception of the PMDA, these regulatory agencies currently accept the transgenic mouse model as an alternative to the MNVT (discussed later).

#### 2.4.1. Current Use of the Monkey Neurovirulence Test

The MNVT is used for lot release testing of other viral vaccines as well, though this varies based on the regulatory agency. The WHO, EMA, and PMDA require the MNVT for measles, mumps, rubella, and varicella vaccines when new master seeds are used. Five consecutive seed lots need to be tested, after which no further testing of the master seed needs to be performed. The US Food and Drug Administration codified requirement for live viral vaccines was removed in 1996 from the code of federal regulations, though it is still used as a guideline. Whether novel vaccines or vaccine lots need to go through the MNVT is determined on a case-by-case basis by the FDA Center for Biologics Evaluation [21]. For novel vaccines, it is generally recommended that they be tested if the wild-type virus is neurovirulent; if it is chimeric virus that has components that are neurovirulent; or if the vaccine was passaged through neuronal cells [21].

Three groups of monkeys are used for inoculation with the agents: one group receives saline or vehicle as a negative control; one group receives the test agent; and the third group receives a positive or benchmark control. Each group of monkeys needs to include at least ten individuals, but regulatory agencies often require twelve or more per group. Monkeys are inoculated in the thalamus, cerebrum, or spinal cord and observed for 30 days. The monkeys are then euthanized, and discriminator regions of the brain are assessed for inflammation. The most commonly used benchmark control is the YFV 17D vaccine, as this is a proven safe and effective vaccine without neurovirulence in patients but causes some inflammatory changes in the CNS of monkeys. Vaccines that cause less inflammation than YFV 17D are considered sufficiently neuroattenuated and deemed safe for clinical use.

#### 2.4.2. Shortcomings of the Monkey Neurovirulence Test

While the MNVT is widely used, it has several shortcomings. Rubin et al. explored these shortcomings specifically regarding neurovirulence testing for mumps vaccines [36], though some of these issues are common for other live viral vaccines. For mumps vaccines, the MNVT only showed non-significant trends towards differences between wild-type and attenuated mumps viruses and failed to detect residual neurovirulence in the Urabe Am9 strain of mumps vaccine [37]. This strain was developed in 1967 based on a Japanese isolate of mumps virus, passaged through chicken and quail cells [38]. The vaccine was widely distributed in Canada, Japan, and Europe until cases of vaccine-associated aseptic meningitis were detected in Canada. More cases were found in Japan and the UK, with an estimated 38–330 cases of aseptic meningitis per 100,000 vaccine recipients [39]. Following these findings, the Urabe Am9 vaccine saw reduced use, and in Japan, mumps was removed as a routine vaccine altogether. Following this policy change, Japan has seen a surge in mumps cases, with up to 1.5 million infections annually [40,41]. The non-translational results of the MNVT led to the scientific community revisiting the value of the MNVT for vaccine safety testing, and in 2005, the International Alliance for Biological Standardization (IABS) released a report on neurovirulence tests for live attenuated vaccines. This workshop report concluded that the monkey neurovirulence test was useful for testing YFV and poliovirus vaccines but was questionably useful for most of the viruses for which it is currently used, including mumps, measles, rubella, influenza, and varicella [21]. The history of the MNVT, these reports, and the development of alternative models in recent years are summarized in Figure 1.

According to ClinicalTrials.gov, over 60 potentially neurovirulent vaccines have had clinical trials since 2000. This list includes live attenuated viral vaccines derived from neurovirulent viruses such as poliovirus, human cytomegalovirus, herpes simplex virus-2, Japanese encephalitis virus, mumps virus, Rift Valley fever virus, varicella virus, and Venezuelan equine encephalitis virus. The list also includes chimeric viral vaccines that used potentially neurovirulent backbones such as vesicular stomatitis virus or measles virus, or vaccines against viruses with neurovirulent potential, such as dengue virus, Japanese encephalitis virus, tick-borne encephalitis virus, and Zika virus. All of these vaccines would need to undergo neurovirulence testing prior to entering clinical trials per regulatory guidelines, and the testing would largely be done through the MNVT. Notably, this list did not include vaccines that did not enter the clinical testing phase, so it is likely that there are more vaccines that have been evaluated using the MNVT.

The use of a single animal model with one endpoint to determine the neurovirulence of viruses from multiple families with different methods of pathogenesis also calls into question the reliance on the MNVT when other models may be just as predictive. The MNVT does not account for many potential mechanisms of neurovirulence. The semi-quantitative scoring system requires significant training for pathologists interpreting the test, and can be a source of variability in testing, leading to lack of reproducibility. Not all neurovirulent viruses cause damage through the same mechanisms. For example, cell death is directly associated with the neurovirulence of alphaviruses [42] but correlates with less virulent strains of rabies virus [43]. Determining the mechanisms of neurovirulence for viral families or even individual viruses will be crucial to finding accurate endpoints for neurovirulence testing. This will help with both vaccine design and in determining endpoints for ensuring adequate neuroattenuation. Molecular endpoints from research into neurodegenerative diseases should be considered in addition to histopathological endpoints, using novel models that adhere to the principles of the 3Rs (Replacement, Reduction and Refinement) for responsible animal testing.

## 3. Animal Models for Testing Vaccine Neurovirulence

Animal models have been historically used to screen vaccines for safety, immunogenicity, and efficacy against a wide range of viruses. While each species has specific advantages and disadvantages, there are some that are consistent across all models. Animal models are superior to in vitro models because all cells of the CNS are present in their homeostatic state; the innate and the adaptive immune response are present; and complex tests such as behavioral screening are possible. Animal models have the major disadvantage of not entirely replicating the human CNS, meaning that viruses and vaccines often need to be screened through multiple species to determine which is the most accurate.

### 3.1. Non-Human Primates

Non-human primates (NHPs) are closely related to humans genetically and have the most similar neuroanatomy and immune system to humans, which would make them seem like the ideal models for determining neurovirulence. The similarity in neuroanatomy to humans allows for experimental results to be more easily translated to potential patient outcomes. NHPs have been used successfully to study mechanisms of viral neurovirulence, including Ebola virus [44], the henipaviruses [45,46], Zika virus [47], alphaviruses [48,49], and influenza virus [50].

Despite the long history of using NHPs in viral research and in screening for neurovirulence, there are numerous disadvantages to their use as a model. NHPs are not ideal disease models for all neurotropic viruses as they do not recapitulate the neurovirulent aspect of some viruses even when inoculated intracranially. Macaques have not been confirmed to be translational models for most of the vaccines tested in the MNVT, including measles, rubella, and varicella. To our knowledge, there have been no studies comparing wild-type and attenuated strains of these viruses in the MNVT to determine if it could discriminate between the two. NHPs show pathology and mortality after wild-type measles infection due to respiratory tract infection [51,52], but no studies in NHPs recapitulate the disease progression in the CNS of patients. Similarly, intracranial inoculation of different mumps strains into NHPs shows no difference between a fully attenuated and an underattenuated vaccine strain, and little difference between wild-type and vaccine strains [37].

Although NHPs are considered the closest model to humans, the differences can affect the translatability of results. Neurons of humans and chimpanzees show small but significant differences in transcriptomics even as progenitor cells [53]. The glial cell to neuron ratios between humans and primates are also significantly different depending on the region of analysis [54]. Astrocytes show morphologic and physiologic differences between species, including between humans and non-chimpanzee primates [55,56]. Smith and Dragunow wrote a review discussing the differences in microglial physiology between humans and other animals [57]. The differences in activity of these cell types could account for the non-translational results seen in the MNVT and could lead to inaccurate results in the future.

NHPs are not ideal for use as a screening tool for vaccines as they are expensive, require extensive animal housing areas, and relatively small numbers of animals are used for studies, leading to less rigorous statistical power. Although at least 30 animals are used per MNVT study, this is still a relatively small number of animals from which to assess human safety. Vaccine neurovirulence does not tend to be very common even when it is reported, such as with the Urabe Am9 vaccine. As such, small numbers of animals may not adequately represent the risk of adverse events in patients. Additionally, there is a major push for reduction in the use of NHPs in research on ethical and humane grounds [58]. Efforts to follow the 3Rs should be encouraged, and MNVT studies require large numbers of animals.

### 3.2. Rodent Models

Rodents are widely used as models for both infectious diseases and neuroscience. Rodents reproduce in large numbers, are genetically well-characterized, require smaller housing than larger animals, and have a wider array of molecular toxicology biomarkers and associated reagents than most other species. Rodents are the most well characterized toxicity model accepted by global health authorities for nonclinical safety testing.

#### 3.2.1. Mice

Mice offer the advantage of having multiple strains with known genetic changes that allow for the study of the roles of specific host genes in a disease process. Genetic knockout is widely performed as a functional screen or to make mice more susceptible to specific disease processes, including viral infection.

Mice are one of the few species that have been accepted as models for vaccine neurovirulence screening. Mouse models of influenza infection using murine-adapted strains are required by the European Medicines Agency to screen for potential neurovirulence in live attenuated influenza vaccines [59]. Importantly, the MNVT is not suggested for screening in this case. A transgenic mouse model expressing the poliovirus receptor is the only accepted in vivo alternative to the MNVT by regulatory agencies for poliovirus. Transgenic mice expressing the poliovirus receptor were developed in 1990 as a model of poliovirus infection [60,61]. This model has shown similar sensitivity to the MNVT for determining residual neurovirulence in poliovirus vaccines [62]. The model has been successful and translational enough to be accepted as a replacement for the MNVT for lot release by many regulatory agencies, though notably not by the PMDA of Japan.

Wild-type strains of mice have been used to study the neurovirulence of a wide range of viruses, as they are susceptible to many different viral infections. Neonatal mice are acutely susceptible to many viruses and have been used extensively to study specific aspects of virus neurovirulence [63,64,65,66]. Their susceptibility makes them a promising model for highly sensitive neurovirulence screening. This was tested using Chimerivax viruses, chimeric flaviviral vaccines based on the YFV 17D backbone. Neonatal mice inoculated intracranially with YFV 17D showed uniform lethality, while those inoculated with varying doses of a chimeric Japanese encephalitis virus vaccine showed reduced mortality. This model, using mortality as an endpoint, was compared to the MNVT, and was found to be more sensitive in detecting potentially neurovirulent changes in the vaccine [67]. While these results are promising, neonatal mice do have some deficiencies as a model. These animals are susceptible to viruses, including live viral vaccines, that normally do not cause disease in humans. While the YFV-17D vaccine is considered safe in humans and non-neurovirulent in the NHP model, it is uniformly lethal in neonatal mice. The high mortality observed with the YFV-17D vaccine in mice makes it problematic as a comparator in mice due to the lack of graded responses. Conversely, infection with wild-type mumps virus causes an abortive infection with no long-lasting consequences or significant clinical signs, even in neonatal mice [68,69].

Transgenic mice are another option for use in screening live viral vaccines for neurovirulence. Transgenic mice can also be used as models of viral infection, through expression of the appropriate receptor or through removal of specific aspects of the immune system. Transgenic mice expressing CD46, the receptor for measles virus, have been used to study the pathogenesis of neurotropic measles virus infections [70,71]. This model has been used to investigate mechanisms of measles neurovirulence and persistence in the CNS but has not been used to differentiate between neurovirulent and non-neurovirulent strains. Mice that are genetically modified to have compromised immune systems have also been suggested as possible models for studying neurovirulence in flaviviruses such as dengue [72,73], offering another option for studying human viruses to which mice are resistant. Although mice can be modified to be susceptible to viral infection, they still have mouse-specific anatomy and physiology with potential differences in susceptibility, tropism, and immune response of the cells to viral infection. Transgenic mice are only useful when there is a known receptor for the virus. In cases where the receptor is unknown, or there are multiple receptors necessary for neurovirulence, a transgenic mouse model is not useful.

#### 3.2.2. Rats

Rats are a widely used model for neuroscience research due to their greater cognitive function than mice. Rats are used less commonly than mice for virology but have been used to study factors of neurovirulence for West Nile virus [74], Borna disease virus [75,76], and influenza virus [77,78,79]. Neonatal rats are a more accurate model of mumps virus neurovirulence than the MNVT. When inoculated intracranially with different strains of mumps virus, neonatal rats develop a degree of cerebral ventricular dilatation that correlates with the neurovirulence of the mumps strain. This dilatation can be used to discriminate between wild-type, underattenuated, and fully attenuated strains of mumps virus [80,81]. This endpoint has been used to further research factors of neurovirulence in mumps, including viral genetic factors [82] and the role of quasispecies [83]. There are no published results of the neonatal rat model to determine if the model can accurately measure neurovirulence using other live viral vaccines such as poliovirus or measles virus.

## 4. Culture-Free Methods

There is currently only one accepted in vitro alternative to the MNVT, specifically for the three types of poliovirus, mutant analysis by PCR and restriction enzyme cleavage (MAPREC). The Sabin live poliovirus vaccines are genetically unstable, and lots are regularly screened for reversion to neurovirulence. MAPREC is used to quantify the population of virus within the vaccine sample containing mutations known to cause neurovirulence [84,85]. These mutations all occur in the 5’ untranslated region of the poliovirus genome, but they vary based on the virus serotype [85]. The total population containing these mutations within a vaccine lot correlates with neurovirulence in the MNVT [86,87]. To quantify the relative population of these mutants in a vaccine sample, the RNA is extracted and transcribed into DNA via reverse transcriptase. The DNA is amplified via PCR with primers that specifically insert restriction sites in the product DNA containing the virulent mutations. The DNA is digested using these restriction enzymes and can then be quantified via radioisotopes [86] or fluorescent dyes [84] to determine the relative population containing those mutations. While this allows for detecting reversion to neurovirulence, it only screens for known, specific mutations and does not account for other mutations that may cause neurovirulence.

Next-generation sequencing has been suggested as an alternative to MAPREC for screening poliovirus vaccine lots due to the reduced number of labs capable of performing the MAPREC [88,89]. While next generation sequencing could be used to screen for live viral vaccines reverting to virulence, it has a similar problem as MAPREC because it is only useful to screen vaccines when the mutations that cause neurovirulence are known and can be detected. To our knowledge, there have not been any studies testing if next generation sequencing can be used to screen other live viral vaccines for neurovirulence. This method would be useful to determine lot-to-lot variability and reversion to virulence in live viral vaccines with known markers of neurovirulence but would not be useful in screening novel vaccines for residual neurovirulence.

## 5. In Vitro Models for Testing Vaccine Neurovirulence

There are no methods accepted by regulatory agencies for screening viral vaccines for neurovirulence using cell culture-based methods. Cell culture-based studies of neurovirulence offer numerous benefits over animal models. There are immortalized, commercially available cell lines derived from humans that can be used to study astrocytes, neurons, microglia, and oligodendrocytes. Individual cell lines allow for investigation of the effects of viral infection on specific cell types. Cell culture models have been used extensively to study mechanisms of neurovirulence in a wide range of viruses. In vitro approaches allow for high-throughput, in-depth screening that is difficult to achieve in animal models. In vitro approaches allow for greater visualization and for assessment of cell function that cannot be accomplished as easily in animal models. Additionally, the cell cultures can be based on human cells from the CNS, allowing for direct comparison without significant genetic differences.

### 5.1. Shortcomings of Immortalized Cell Lines in Neurovirulence Assays

Most in vitro methods suffer from a lack of translatability. The CNS is a complex web of cell interactions, and the interactions between cells can affect the susceptibility and responses to viral infections. Even isolated primary cells from the CNS act differently in vitro than in vivo, likely due in large part to the removal of interactions with the other cells of the CNS. For instance, primary astrocytes in a monoculture take on an activated phenotype that is more akin to an inflammatory state than they do when in the healthy CNS [90], in part due to contact with serum proteins necessary for cell culture [91,92]. Though there have been advances in developing models to create monocultures with a quiescent phenotype [93], cultures of cells from the CNS are more translational when cultured with multiple different cell types.

The source of the cells used in culture can have a major effect on the results of experiments done to study viruses as well. Primary cells can be used but are difficult to acquire in the cases of human cells, are often difficult to culture, and reduce the volume of tests that can be performed. Immortalized cells can be used but have significant changes in activity. These changes can include genetic changes, transcriptional divergence, and changes in protein expression [94,95], all of which can affect the cells’ interactions with viruses.

### 5.2. Human Induced Pluripotent Stem Cells

Human induced pluripotent stem cells (hiPSCs) are an emerging technology that have been developed for multiple organs. To create hiPSCs, fibroblasts are harvested from patients, then induced to become stem cells, which can then be differentiated into human cells in culture. The stem cells can be maintained in culture precluding the need for repeated harvesting of stem cells. This process allows for the study of primary human cells without needing to collect them from patients and allows for larger scale studies as the populations can be grown in vitro. The opportunities for hiPSC-derived neural cell lines in studying virus-CNS interactions have been reviewed by Harschnitz and Studer [96]. Notable conclusions from that paper that should be considered are that the hiPSC models allow for scalability and flexibility in models; that they allow for high-throughput screening; and that they more closely mimic the immune responses of cells of the central nervous system than other models. All of these factors indicate that they would be potential models for use in determining the neurovirulence of live viral vaccines.

The first factor in determining if hiPSC-derived cell culture would be a good model for neurovirulence testing is determining how closely they match in physiology to their counterparts in vivo. HiPSC-derived models show a high level of translatability to the human brain. Co-cultures derived from hiPSC show activity in vitro that indicate biological activity mimicking neurons. Neurons derived from hiPSCs express ligand-gated channels that are functional [97], generate action potentials, form functional synapses [98] with specific presynaptic and postsynaptic specializations, express surface markers consistent with human neurons [99], release neurotransmitters [100,101,102], and maintain long-term spontaneous activity which allows for drug-screening ability [103]. The neurons show increased translatability when co-cultured with astrocytes, forming neural networks in vitro [104]. The use of hiPSC-derived neurons to study neurodegenerative diseases was recently reviewed by Devine and Patani [105]. This review emphasizes the translatability of hiPSCs for studying human disease and how they can be used for drug screening. Many of the same workflows could be applied to screening vaccine candidates, and the high-throughput capability of hiPSC-derived cells would be beneficial to vaccine development.

Co-cultures of hiPSCs better represent the CNS in a homeostatic state as the different cell types communicate with each other and alter each other’s activity. Co-cultures of neurons and astrocytes are the most commonly used model, but the introduction of microglia into the system is being optimized. Recent work shows success in defining microglia that are very similar to microglia isolated from human brain, and these have been used in a tri-culture system to investigate neurodegenerative diseases [106]. Transcriptionally, microglia derived from hiPSC are similar to their primary isolated counterparts, and act similarly in vitro and when transplanted into mice [107,108]. Microglia continue to be refined and have begun to approach the phenotype and RNA profile of isolated microglia from human brains [109]. One group developed a method to derive microglia that led to highly pure microglia with a similar phenotype to primary microglia [106]. This model recapitulated neuroinflammatory responses to stimulation via lipopolysaccharide or APP mutant cells. The astrocytes and microglia in these models enter an inflammatory loop where C3 is produced by both astrocytes and microglia, indicating a role of both cells in neuroinflammation. These findings indicate that having all three cell types is crucial to accurately model the human CNS in vitro.

#### 5.2.1. HiPSC-Derived Cell Culture as a Model for Viral Neurovirulence

More pertinent to vaccine safety evaluation, hiPSCs have been used to study neurotropic viruses including Zika virus, human immunodeficiency virus (HIV), and human cytomegalovirus (HCMV), herpes simplex virus-1 (HSV-1) [110], and measles virus [111]. Research using hiPSCs to study Zika virus showed astrocytes are infected with Zika virus [112], and develop mitochondrial damage, leading to production of free radicals and DNA damage [113]. HiPSCs were also used to determine that Zika virus infects neural progenitor cells, inducing apoptosis in this cell population [114,115,116], which is presumed to be the mechanism for the neurological damage Zika virus infection causes [117]. An hiPSC-based system was used to study the tropism of SARS-CoV-2, using monocultures and a tri-culture system. The tri-culture system was not fully explored and characterized, but a separate monoculture was used to determine the virus’ tropism for choroid plexus cells [118]. For HIV, an hiPSC system was used to show that HIV causes increased EIF2 signaling, indicating ongoing inflammation [109]. The findings from that study mirror findings using post-mortem tissues from HIV patients [119,120], indicating a high level of translatability. Although there are no published studies using hiPSC-derived cells to model established, neuroattenuated live viral vaccines, the studies modeling viral infections of the CNS indicate a high level of translatability.

One important consideration for using the hiPSC-derived cell culture model is determining what endpoints should be used to detect neurovirulence. There have been no studies comparing wild-type and attenuated viral infections in this model, so initial work would need to focus on basic discovery. Measuring viral replication would be a possibility, but some live attenuated viral vaccines can grow in neuronal cultures with little to no difference compared to the wild-type viruses [121]. Assessment of cell viability, including cell death, would be the simplest option. This could be done using a TUNEL assay. Another would be assessing functional activity of the co-cultures, including electrical activity. Electrical activity in the neurons can be an indication of neuronal dysfunction without obvious cell death. Like animal models, there may need to multiple endpoints assessed to measure neurovirulence, and the endpoints likely will be different for different viral families or species.

#### 5.2.2. Potential Drawbacks of hiPSC-Derived Cell Culture

While the use of hiPSC-derived cells does open avenues for use in vaccine development and testing, there are potential shortcomings of the model. One major disadvantage of the hiPSC approach compared to animal models is that it cannot currently reproduce the heterogeneity of the CNS. Different protocols are required to differentiate neurons into neurons with specific region and neurotransmitter identities [98,101,122,123]. Although different general cell types can be produced, the regions of the brain have unique innate immune responses and susceptibility to infection [124,125,126]. Although neurons can be produced, they are generally of only a few phenotypes, and without knowing exactly which cells are affected, the model may not accurately reflect the neurons that the virus normally infects.

Early hiPSC-derived cultures more closely approximated the fetal or neonatal nervous system rather than adult. This may be an asset as they may be more sensitive to viral infection and thus a more sensitive method for screening live viral vaccines. There have been advances in producing more mature cultures that mimic adult or aged nervous systems, particularly with a focus on neurodegenerative diseases [127]. Research has been performed using progerin to artificially age hiPSC-derived neurons and develop adult-stage neurons [128]. Further work and discussion with regulatory agencies would need to be conducted to determine what form of neurons would be best suited for screening for residual neurovirulence.

The heterogeneity in methods for differentiating cells of the CNS in these cultures can lead to variability in outcomes. The methods used to measure the translatability of the system vary as well, confounding the ability to make direct comparisons between approaches. Engle et al. did a systematic review of small-molecule compound screens using neurons derived from hiPSCs and found significant differences between research practices in the amount of technical detail, relevance of the assays, and reproducibility of the techniques between studies [129]. The quality and validity of the studies do appear to be improving over time, but it would be important to rigorously characterize the cells produced by any particular method before and after infection to ensure translatability of the results.

## 6. Future Directions

Live viral vaccines continue as a strong option for global vaccine development. They generally require fewer doses than adjuvanted or mRNA vaccines and often require less stringent handling conditions. These factors make them amenable to use in areas without the infrastructure to track vaccination status of a population or to maintain vaccine cold chains. However, the risk of reversion to virulence or lack of initial attenuation needs to be considered for any live viral vaccine developed, unlike for subunit or inactivated vaccines. Novel models are needed for testing new live attenuated vaccines. The rat neurovirulence model is one option, as are the mouse models for flaviviruses. More work needs to be done to examine these models for use with other viruses.

The different models presented in this review, as well as the pros and cons of each, are listed in Figure 2. The MNVT has been historically used for vaccine testing for almost 80 years and has been shown to be translational for YFV and poliovirus vaccines. However, given its shortcomings, other models need to be investigated. The introduction of novel hiPSC-derived cultures could open the way for better screening tools that allow for more high-throughput screening, while rodent models offer more sensitivity for certain viruses. It is likely that no single model will be able to predict neurovirulence for all types of live viral vaccine and using knowledge of the individual virus’ pathogenesis will help direct toward the appropriate model. Additionally, the endpoints used in each model and potentially for each virus may be different.

Through natural infection, viruses can acquire mutations that lead to neurovirulence. These same mutations can make the virus more difficult to isolate and use in a laboratory setting. For example, measles virus requires significant mutations to become neurovirulent, and these mutations reduce its ability to form viral particles. Therefore, the mutant virus cannot be isolated for use in a neurovirulence test. Using strains of these viruses without a neurovirulent phenotype as positive controls for neurovirulence testing may lead to inaccurate results. Whenever possible, strains with the neurovirulent phenotype should be used. When this is not possible, alternative positive controls should be considered. Recently, research using measles virus including select mutations associated with neurovirulence that could be isolated was performed using ex vivo mouse brains and hiPSC-derived neural cells [111]. 

Updating the safety standards for live viral vaccines is an enormous effort and will require collaboration between pharmaceutical companies, regulatory agencies, and academic researchers. The models mentioned in this review will need to be qualified to determine their sensitivity and specificity in detecting neurovirulence using different viruses; endpoints will need to be determined in each model; and results will need to be proven reproducible at multiple sites. Merck has been collaborating with the Critical Path Institute (C-Path) to establish a consortium aimed at addressing this need. Scientists from multiple vaccine manufacturers, governmental agencies, and academic institutions have begun work studying possible new models for screening live viral vaccines for neurovirulence. The goal of this effort will be to qualify new models that will be accepted by all major health authorities including the FDA, EMA, PMDA and the WHO.

## 7. Conclusions

There has only been one case where the MNVT failed to predict neurovirulence, the Urabe Am9 mumps vaccine, but this failure has had massive consequences. The neurovirulence associated with this mumps vaccine strain led to the removal of any mumps vaccine from routine childhood immunization schedule in Japan. Similar outcomes for other viruses could lead to significant public hesitancy to widespread vaccination. Given that it takes a significant amount of time and resources to ensure neurovirulence has been eliminated from a candidate vaccine, more sensitive derisking models need to be pursued. Despite never having been fully qualified for most of the viruses used, the MNVT continues to be the gold standard for neurovirulence testing. Rodent neurovirulence models are an alternative that shows translatability, but work needs to be done to determine if this is virus-specific, and to determine the sensitivity, specificity and reproducibility of neonatal rodents as models for viral neurovirulence.

In vitro methods, particularly hiPSC-derived models, appear to be a promising lead in studying viral neurovirulence, which could easily lead to high-throughput screening for candidate vaccines. The biggest hurdle this model faces now is a shortage of data. There are no published studies investigating the effects of live viral vaccines in hiPSC-based systems, nor are there studies comparing candidate vaccines to their wild-type counterparts. More work needs to be done to determine how closely hiPSC-derived models compare to the human CNS from a viral infection standpoint, and to determine how much the lack of an adaptive immune response affects this susceptibility.

There is an ongoing need for the development of safe, efficacious vaccines, and screening for neurovirulence will play a role in the success or failure of many of these vaccines. To quickly and efficiently respond to future viral outbreaks, it will be crucial to develop better models of viral neurovirulence.

## Figures and Tables

**Figure 1 vaccines-09-00710-f001:**
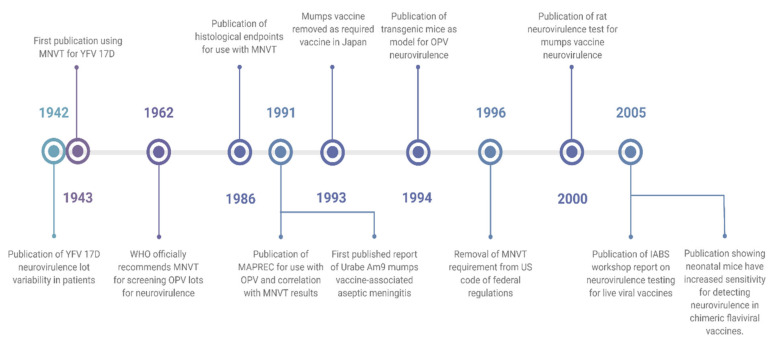
Timeline from 1942 until 2005 showing the progression of live viral vaccine neurovirulence testing. Created with BioRender.com.

**Figure 2 vaccines-09-00710-f002:**
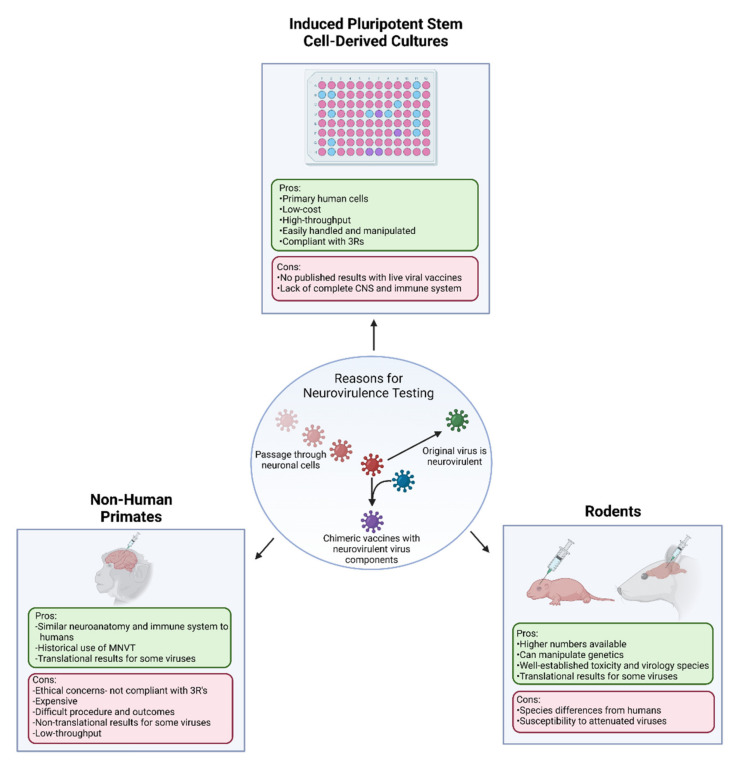
Overview of live viral vaccine neurovirulence testing. Live viral vaccines need to be tested for neurovirulence when the parent virus is considered neurovirulent, when components of a chimeric or viral vectored vaccine are from neurovirulent viruses, or when the vaccine was derived by passage through cells of neuronal origin. The currently used method, the monkey neurovirulence test, uses intracranial inoculation. Adult or neonatal rodents inoculated intracranially have been used as a model for mumps vaccine neurovirulence testing and are under investigation as an alternative model. Human induced pluripotent stem cells (hiPSCs) have not been used to study live viral vaccine neurovirulence yet but have been used to study virus neurovirulence. This model presents a promising potential model that will be investigated due to its translational and high-throughput capabilities. Created with BioRender.com.

## Data Availability

Data sharing not applicable.

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
