# Peer review of "Live Viral Vaccine Neurovirulence Screening: Current and Future Models"

_vaccines, 2021, doi:10.3390/vaccines9070710_

Round 1

Reviewer 1 Report

In the review article “Live Viral Vaccine Neurovirulence Screening: Current and Future Models” by Fulton and Bailey, the current state of models used to study neurovirulence and screen live viral vaccines for neurovirulence attenuation are discussed. Neurological effects are common with many emerging viral diseases and the live viral vaccines that are developed in an attempt to control disease caused by these viruses must be tested to ensure there are no residual effects. In this review, the authors cover the MNVT test, including its historical use and its pitfalls, as well as new potential alternative in vitro and in vivo methods. The authors provide a general review of neurinvasion vs neurovirulence; live viral vaccines; attenuation of neurovirulence in live viral vaccines; monkey neurovirulence test MNVT; animal models for testing vaccine neurovirulence; culture-free methods; and in vitro methods for testing vaccine neurovirulence.

Overall, the review is very well-written. The authors clearly present the rationale for needing new and additional screening models, given some of the pitfalls of the somewhat costly and often variable MNVT model. The authors were good at citing and highlighting other review articles that cover topics in greater depth than in their review. The authors also thoroughly detailed the benefits and limitations of each potential alternative model discussed, as well as some of the technical background of their development and use. The regulatory considerations were also appreciated, as this knowledge will be important to consider moving forward. 

More specific issues/points that would improve the review:

  1. The figure and table included are great, but I felt that additional figures could be included to visually present some of the subject matter. For instance, it might be useful to make a figure comparing the MNVT and alternative methods as an efficient visual snapshot to convey pros/cons and major considerations. 
  2. There were a few sections that would benefit from better organization, including additional ‘sub-sections’ with section headers. For instance, “The Existing Test for Neurovirulence: The MNVT” section could be broken into sections that cover the test development and procedure, and another one that highlights the problems that have emerged (Urabe Am9) and the rationale to pursue alternative tests, etc. The same is true for In Vitro models section. It could be broken into subsections for each model type similar to how the Animal models section is organized. As written, the length of this section distracts from its importance, and as such it would greatly benefit from being broken down into relevant sub-sections with descriptive sub-headers.
  3. Neurovirulence appears to be associated with significant mutation to the viral genome in some viruses, yet the tests for neurovirulence involve live viral vaccines with wild-type strains. Do the new/alternative models consider this? If so, how?
  4. Section 3.2 Rodent Models followed by section 3.4 Mice---is there a section 3.3 missing?
  5. Page 9, Line 398: “The” should not be capitalized, and this sentence seems fragmented or like it is an inadvertent combination of two sentences.

Author Response

We appreciate the constructive criticism regarding our review submission, “Live Viral Vaccine Neurovirulence Screening: Current and Fu-ture Models.” We have integrated the changes into the manuscript. Addressing the specific changes requested:

  1. The figure and table included are great, but I felt that additional figures could be included to visually present some of the subject matter. For instance, it might be useful to make a figure comparing the MNVT and alternative methods as an efficient visual snapshot to convey pros/cons and major considerations.
    1. Figure 2 was added and replaced the table, indicating when neurovirulence testing needs to be performed for live viral vaccines, and the general pros and cons of each of the models.
  2. There were a few sections that would benefit from better organization, including additional ‘sub-sections’ with section headers. For instance, “The Existing Test for Neurovirulence: The MNVT” section could be broken into sections that cover the test development and procedure, and another one that highlights the problems that have emerged (Urabe Am9) and the rationale to pursue alternative tests, etc. The same is true for In Vitro models section. It could be broken into subsections for each model type similar to how the Animal models section is organized. As written, the length of this section distracts from its importance, and as such it would greatly benefit from being broken down into relevant sub-sections with descriptive sub-headers.
    1. We agree with this assessment, and added subheaders into the MNVT and hiPSC-derived culture sections of the paper.
  3. Neurovirulence appears to be associated with significant mutation to the viral genome in some viruses, yet the tests for neurovirulence involve live viral vaccines with wild-type strains. Do the new/alternative models consider this? If so, how?
    1. This is a good point. We clarified that this is primarily the case for measles virus in the neurovirulence section and that this makes it difficult to study. The other testing methods may or may not do a better job of screening for these outcomes, so we added a paragraph in the “future directions” section (lines 556-566) to mention recent research that we feel is relevant.
  4. Section 3.2 Rodent Models followed by section 3.4 Mice---is there a section 3.3 missing?
    1. This was a typo and has been fixed.
  5. Page 9, Line 398: “The” should not be capitalized, and this sentence seems fragmented or like it is an inadvertent combination of two sentences.
    1. Thank you for catching this. We removed that fragment.

Reviewer 2 Report

This review discusses about different  live viral vaccine and to check its virulence in different animal model. This is an important review regarding viral vaccine and worthful for scientists. 

Could author summarise this review in some schematic diagram, so it would be easy for audience to understand?

Besides, can authors give some information about live vaccines in practice if any, which people are using it now. 

Author Response

We appreciate the constructive criticism regarding our review submission, “Live Viral Vaccine Neurovirulence Screening: Current and Fu-ture Models.” We have integrated the changes into the manuscript. Addressing the specific changes requested:

  1. Could author summarise this review in some schematic diagram, so it would be easy for audience to understand?
    1. We have included a second figure that addresses this, and includes reasons why live viral vaccines may need to be tested for neurovirulence and the different methods we have proposed.
  2. Besides, can authors give some information about live vaccines in practice if any, which people are using it now. 
    1. Yes, we added a section in the introduction on live viral vaccines that are currently in use and recent developments (lines 95-102).